# In Vitro Antiproliferative Activity of *Ptaeroxylon obliquum* Leaf Extracts, Fractions and Isolated Compounds on Several Cancer Cell Lines

Edward T. Khunoana [1], Jacobus N. Eloff [1], Thanyani E. Ramadwa [2], Sanah M. Nkadimeng [1,2], Mamoalosi A. Selepe [3] and Lyndy J. McGaw [1,*]

1 Phytomedicine Programme, Department of Paraclinical Sciences, Faculty of Veterinary Science, University of Pretoria, Private Bag X04, Pretoria 0110, South Africa
2 Department of Life and Consumer Sciences, College of Agriculture and Environmental Sciences, Florida Campus, University of South Africa, Private Bag X6, Florida 1710, South Africa
3 Department of Chemistry, University of Pretoria, Lynnwood Rd., Pretoria 0002, South Africa
* Correspondence: lyndy.mcgaw@up.ac.za; Tel.: +27-(0)12-529-8351

**Abstract:** Several cancers are induced by microbial infections or chronic inflammation. *Ptaeroxylon obliquum* is traditionally used to treat various infections characterized by inflammation. The in vitro antiproliferative and antioxidant activity of *P. obliquum* leaf extracts, fractions and isolated compounds were determined. Antiproliferative activity was assessed against normal Vero cells, and several cancerous human cells, including human breast cancer (MCF-7), hepatocarcinoma (HepG2), lung adenocarcinoma (A549) and human cervical cancer cells (HeLa) using a colorimetric tetrazolium bromide assay. Radical scavenging activity was tested using the 2,2-diphenyl-1-instrpicrylhydrazyl (DPPH) and 2,2'-azino-bis-3-ethylbenzothiazoline-6-sulfonic acid (ABTS) assays. Obliquumol, *O*-methylalloptaeroxylin and a mixture of lupeol and β-amyrin were isolated from the chloroform fraction using silica gel open column chromatography. Acetone extracts were toxic to HepG2 cells with IC$_{50}$ values from 8 to 200 μg/mL but were less toxic to other cells with selectivity index as high as 14. Aqueous extracts and fractions were non-toxic at concentrations tested against all the cell lines (IC$_{50}$ > 100 μg/mL). Isolated compounds had IC$_{50}$ values ranging from 52 to 539 μg/mL and 189 to 247 μg/mL against HepG2 and HeLa cells, respectively. Light microscopy showing changes in HepG2 and HeLa cell morphology supported the cytotoxicity of the acetone extracts. Water extracts scavenged ABTS and DPPH radicals with IC$_{50}$ values as low as 29.06 μg/mL and 43.4 μg/mL. *P. obliquum* extracts may be useful as sources of anticancer therapy, as they have selective cytotoxicity against cancer cell lines.

**Keywords:** *Ptaeroxylon obliquum*; cancer; antiproliferative; Vero; HepG2; HeLa

## 1. Introduction

Cancer is an unusual formation of cells caused by various changes in gene expression leading to dysregulated balance of cell proliferation and cell death [1]. This finally leads to a population of cells that invade tissues and then metastasize to distant sites, causing significant morbidity, and if not treated, mortality [2]. Cancer is initiated by mutations in DNA that activate oncogenes and inactivate tumour suppressors; it thrives when changes occur in the host metabolism and cell structure [3,4]. The development process of normal, healthy cells turning into cancer cells is termed carcinogenesis and this process takes place in three stages, namely initiation, promotion and progression [2].

One of the leading causes of death globally is cancer, which has a 20% mortality rate estimated at 18.1 million under non-communicable diseases, with a 33% increase in newly diagnosed cases estimated at 9.6 million between 2015 and 2018 [5–7]. The World Health Organization (WHO) projections indicate that by 2040, these numbers will have increased

to 29.5 million of new cancer diagnoses and 16.5 million cancer-related deaths yearly across the globe [5–7]. In South Africa, cancer is emerging as a critical public health problem with an estimated 107,467 new reported cancer cases and a total of 57,373 deaths occurring in 2018 [8,9]. The incidence rate of breast cancer is the highest (14,097 cases, 13.1%), followed by cervical (12,983 cases, 12.1%) and prostate cancers (12,452 cases, 11.6%) and the incidence of lung cancer was reported at 7.7%, ranking fourth among all the cancers, which gives it the highest mortality rate of 13.5%. Lung and prostate cancers are the most predominant types in males while breast and cervical cancers are the predominant types in females [9,10].

Treatment options for cancer include chemotherapy, radiation, hormone and gene therapy; however, they all have various negative side effects such as fatigue, nausea, vomiting, weight loss and bleeding [11]. Chemotherapy is one of the most commonly used anticancer treatments, but some cancers are resistant to cytotoxic/chemotherapeutic agents, which poses a major threat to anticancer therapy. The limitation in the efficacy of the therapeutic agents leads to non-satisfactory treatment outcomes and eventually death [12]. Cancer cells may be resistant by evading potential apoptotic mechanisms, such as down-regulated pro-apoptotic signals, up-regulated anti-apoptotic signals, and faulty apoptosis initiation and implementation.

There is a need to seek alternative anticancer therapeutics such as phytochemicals, as there is increasing evidence that suggest they could exhibit anticancer effects. Scientific evidence suggests that phytochemicals have substantial anticancer potential that may be considered for drug development [13]. Modern medicine has embraced plants used in traditional medicine as potential leads for the development of therapeutic drugs. South Africa has a large variety of plant species, which are yet to be investigated for their potential in treating cancer. Natural products obtained from plants, marine organisms and microorganisms, account for approximately 60% of the currently used anticancer agents, of which about 25% have been sourced from plants [14].

Approximately 20% of cancers are induced by chronic inflammation or other infections [15]. During chronic inflammation, reactive oxygen/nitrogen species (ROS/RNS) are produced from inflammatory cells and epithelial cells [15]. ROS/RNS cause DNA damage in organs during inflammation, leading to cancer development. A biological system is under oxidative stress when there is an imbalance between the synthesis and expression of reactive oxygen species and its ability to quickly detoxify the reactive intermediates or to repair the resultant damage re [1]. Oxidative stress is closely linked to every aspect of cancer, including prevention, tumor development and therapy [16]. Numerous studies have shown that oxidative stress and human pathophysiological disorders may be fundamentally related [17]. Specifically, it is well recognized that oxidative stress affects the DNA molecule, changes signaling pathways, and controls the development of a variety of cancers, including those of the breast, lung, liver, colon, prostate, ovary and brain [13,18].

*Ptaeroxylon obliquum* (Thunb.) Radlk. (Rutaceae), also known as the sneezewood tree, is used in traditional medicine to treat many infections, including inflammatory-related diseases in South Africa [19]. Animal and human illnesses have long been treated using sneezewood. In Portugal, the bark is used to cure fevers, arthritis, and rheumatism [20]. The Xhosa people snuff the powdered bark material as a recreational and therapeutic remedy for headache relief [21]. The plant's wood is used as a therapy for anthrax, rheumatism, heart conditions, lupus, warts, sinusitis, the treatment of individuals who experience fits, as a tick repellent for cattle, and in ritual sacrifices to ancestor spirits. Wooden pegs are frequently used as lighting protection [22]. To keep moths and other insects away from cupboards, wood pieces are still used. Its ability to deter insects made it a popular wood for bedsteads [23].

Therefore, the present study was aimed at investigating the antiproliferative and antioxidant properties of *P. obliquum* extracts, fractions, and isolated compounds against different cancer cell lines. Leaf samples were gathered from a variety of geographic regions since regional diversity may have some effects on the concentration of bioactive compounds in plants of the same species [24].

## 2. Materials and Methods

### 2.1. Plant Collection

*P. obliquum* leaves used in this study were collected during the summer of 2019 from trees growing at the Hatfield Campus (University of Pretoria), National Botanical Gardens of the South African National Biodiversity Institute (SANBI) in Pretoria, the Lowveld Botanic Gardens (Nelspruit, Mpumalanga) and Walter Sisulu National Botanical Garden (Roodepoort). Leaves were collected in open weave nylon bags, dried in the shade at room temperature, and powdered using a grinder. The powders were stored in closed containers in the dark until needed. Voucher specimens (PRU130509, PRU130510, PRU130628 and PRU130627) were prepared and kept at the HGWJ Schweickerdt Herbarium, University of Pretoria. Bulk plant material for isolation could only be collected at SANBI and Nelspruit due to limited availability.

### 2.2. Preparation of Extracts

Exactly 5 g of the powder was extracted using 50 mL acetone and distilled water (hot and cold) separately. The mixture was placed in an airtight container on a shaker and left for 24 h, after which the supernatant was filtered through Whatman No. 1 filter paper and placed into a pre-weighed honey jar. The process was repeated three times for each solvent. Then the supernatants for each solvent were combined in a single pre-weighed honey jar and dried under a stream of cold air.

### 2.3. Fractionation and Isolation of Bioactive Compounds from P. obliquum

*P. obliquum* leaf powder (500 g) from SANBI and Nelspruit were extracted separately with 5L of acetone and vigorously shaken for 8 h on a Labotec shaking machine. A Büchner funnel was used to filter the supernatant through Whatman No. 1 filter paper, and a Büchi Rotavapor R-114 (Labotec) was used to evaporate the solvent under vacuum. The concentrated extract was then transferred to a pre-weighed beaker. The same technique was carried out again on the plant material. The extracted mass was then measured after the extract had been allowed to dry at room temperature, yielding 36.6 g (Nelspruit) and 42.17 g (SANBI). The solvent–solvent extraction/fractionation of plant extracts protocol was developed by the National Cancer Institute, modified by eliminating the carbon tetrachloride extraction step. Five solvent–solvent fractions containing compounds with different polarities from *P. obliquum* acetone leaf extract were then obtained [25,26]. To obtain the initial $CHCl_3$ and $H_2O$ fractions, the acetone extract was reconstituted in 500 mL of $CHCl_3$: $H_2O$ (1:1) in a separatory funnel and the two layers were partitioned. The $H_2O$ and *n*-BuOH fractions were then obtained by combining the $H_2O$ fraction with an equivalent volume of *n*-BuOH. The *n*-hexane fraction was obtained by extracting the initial $CHCl_3$ fraction with an equal volume of *n*-hexane and a 10% $H_2O$-methanol mixture after the initial $CHCl_3$ fraction had been dried in a vacuum rotary evaporator. To create the 35% $H_2O$-MeOH fraction and the $CHCl_3$ fraction, the 10% $H_2O$-MeOH fraction was first diluted to 35% $H_2O$-MeOH. Therefore, five fractions were obtained: $H_2O$, n-BuOH, 35% $H_2O$-MeOH, $CHCl_3$, and *n*-hexane fractions.

Column chromatography was used to isolate the bioactive chemicals from the SANBI $CHCl_3$ fraction, with silica gel serving as the stationary phase. A uniform slurry made from about 1000 g of silica gel (Merck) and *n*-hexane was then loaded into a glass column with the dimensions of 40 cm in height and 4.5 cm in diameter. The dried $CHCl_3$ fraction (36.47 g) was combined with 50 g of silica gel prior to the addition of 100 mL of acetone, before drying for approximately 2–3 h at room temperature. The column bed was then overlaid with the dried $CHCl_3$ fraction. *n*-Hexane with ethyl acetate (EtOAc) was added in increasing percentages (5%) to 100% to yield various fractions that were collected.

After the fractions had dried, white precipitates were observed in some honey jars from 70% to 60% *n*-hexane fractions and were then washed with acetone to remove impurities and yielded 50 mg of pure white crystals of obliquumol.

By constantly purifying the dried fractions collected at *n*-hexane concentrations ranging from 95 to 85% with EtOAc, a 140 mg combination of a mixture of lupeol and β-amyrin was obtained.

Fractions collected from 30–5% hexane were combined, since they contained similar compounds based on TLC finger printing. The 1.2 g yield from the combined fractions from the first column was dissolved in acetone, combined with 1 g of silica gel, and allowed to dry at room temperature. The sample was then deposited into a silica gel bed with dimensions of 40 cm in height by 2.5 cm in diameter, and it was eluted using a solution of 70% to 30% EtOAc and *n*-hexane. About 50 mg of *O*-methylalloptaeroxylin was then isolated. Nuclear magnetic resonance (NMR) spectroscopy and liquid chromatography–mass spectrometry (LC-MS) were used to analyse the samples and the structures of the isolated compounds were elucidated. The 1-dimensional (1D) NMR ($^1$H, $^{13}$C, and dept-135) and 2-dimensional (2D) NMR (COSY, HMBC, HSQC, and NOESY) spectra were used. The data was also compared with the literature to conclusively interpret the structures.

### 2.4. In Vitro Cytotoxicity Assay

### 2.4.1. Cell Cultures

Human liver hepatocarcinoma cells (HepG2), human breast adenocarcinoma cells (MCF-7), human cervical cancer cells (HeLa), human lung adenocarcinoma (A549) and African green monkey kidney cells (Vero) were cultured in Dulbecco's Modified Eagle Medium (DMEM) and minimum essential medium (MEM), supplemented with 10% foetal bovine serum (FBS) and 1% gentamicin solution. The cells were grown in 5% $CO_2$ at 37 °C in a humidified atmosphere.

### 2.4.2. The 3-(4,5-Dimethyltetrazolium Bromide) (MTT) Reduction Assay

Cytotoxicity was evaluated using the 3-(4,5-dimethylthiazol-2-yl)-2,5-diphenyl tetrazolium bromide (MTT) reduction assay described by [27], with minor modifications. Cells (100 μL) were seeded at $1 \times 10^5$ cells/mL density in 96-well microtitre plates and incubated at 37 °C in a 5% $CO_2$ incubator for 24 h to allow attachment of the cells. Extracts and fractions were resuspended in acetone to 100 mg/mL and the compounds were resuspended in DMSO to 20 μg/mL, then serial dilutions of the samples were prepared in DMEM (10% FBS and 1% gentamicin solution). After the incubation period, 100 μL of each sample were added to the wells containing cells. Doxorubicin was used as a positive control. Negative controls with the same volume of solvents were also included, and the plates were further incubated for 48 h in a $CO_2$ incubator. Following incubation, medium in each well was removed from the cells, which were then washed with approximately 150 μL of PBS. The PBS was aspirated, and fresh medium (200 μL) was added to all the wells. Finally, 30 μL of MTT (5 mg/mL in PBS) was added to each well and the microtitre plates were incubated at 37°C for 4 h. Following 4 h incubation, the medium was aspirated from the wells, and 50 μL of DMSO added to solubilize the resulting formazan crystals. The absorbance was read using a microplate reader (Bio-Tek Synergy, Instruments Inc, Santa Clara, CA, USA) at 570 nm with a reference wavelength of 630 nm. The $IC_{50}$ values were calculated as the concentration of the tested samples, resulting in a 50% reduction of absorbance compared to untreated cells. The relative safety of each sample was assessed using the selectivity index, which was calculated as follows:

% viability = (absorbance of sample treated cells/absorbance of control cells) × 100.

All experiments were performed in triplicate and mean values were calculated.

### 2.4.3. Selectivity Index (SI)

The selectivity index (SI) indicates the degree of cytotoxic selectivity of the tested sample against cancer cells versus normal cells (Vero) and was calculated by dividing the $IC_{50}$ of the tested sample in normal cells by the $IC_{50}$ of cancer cells.

The SI values were calculated by applying the formula:

$$SI = IC_{50} \text{ normal cell}/IC_{50} \text{ cancer cell.}$$

### 2.5. Morphological Study

Morphological alteration of HepG2 and HeLa cell lines after exposure to test substances was assessed under the microscope. The cells were seeded at a density of $1 \times 10^5$ cells/mL in 5 mL medium in a 25 cm³ flask. After 24 h, the medium was removed and replaced with new medium. Thereafter, HepG2, and HeLa cell lines were treated with 100 μg/mL, 50 μg/mL and 25 μg/mL of the acetone leaf extracts for 48 h. After the treatment, the images were captured at 100× magnification, using a phase contrast inverted microscope (Nikon Eclipse Ti Optical Co., Ltd., Tokyo, Japan). Doxorubicin (12 μg/mL) was used as a positive control, while the untreated cells were the negative control. The effect of *P. obliquum* acetone leaf extracts from two different geographic locations on morphological changes of HepG2 and HeLa cells was assessed and photographed.

### 2.6. Antioxidant Activity of P. obliquum Extracts and Fractions

### 2.6.1. Quantitative 1,1-Diphenyl-2-Picrylhydrazyl (DPPH) Free Radical Scavenging Method

The effects that the extracts and fractions had on the DPPH radical were determined using a DPPH radical scavenging assay as described by [28], with slight modifications. The extracts and fractions were re-dissolved to a concentration of 10 mg/mL in methanol. First, the DPPH solution's optical density (OD) was calibrated at 517 nm to a range of 0.9 to 1.00. Then, 160 μg/mL of the DPPH solution were added to 40 μg/mL of various crude extracts and fractions at various concentrations (3.125–200 μg/mL). Using a microplate reader, the mixture was incubated in the dark for 30 min to measure the absorbance at 517 nm (Bio-Tek Epoch spectrophotometer, Instruments Inc, Santa Clara, CA, USA). Higher free radical scavenging activity was shown by the solution's lower absorbance. Ascorbic acid and trolox were used as positive controls, with methanol serving as the negative control. The experiment was repeated three times. The percentage inhibition was calculated as:

$$\% \ inhibition = 100 - \left( \left( \frac{\text{Sample} - \text{Control}}{\text{DPPH}} \right) \times 100 \right)$$

The $IC_{50}$ is the concentration of the sample that can inhibit 50% of the radicals in the DPPH. The lower the $IC_{50}$ value of the samples, the more effective is the antioxidant activity [28].

### 2.6.2. ABTS Free Radical Scavenging Method

The 2,2′-azinobis(3-ethylbenzothiazoline-6-sulfonate) radical cation (ABTS+) assay was carried out on all the extracts and fractions. The ABTS stock solution was made by combining a 2.45 mM potassium persulfate solution with a 7 mM ABTS in methanol solution, and letting the combination sit at room temperature in a dark area for 12 to 16 h [29]. The extracts and fractions were re-suspended in methanol to a concentration of 10 mg/mL. A volume of 40 μL of the samples were diluted to 50% with methanol and then serially diluted in a 96-well microplate. The absorbance of the stock ABTS solution was measured using a spectrophotometer at 734 nm to an absorbance of 0.7–1.0, then 160 μL was added to all the wells of the microplate. The microplates were then incubated in the dark at room temperature for 6 min, and then the absorbance was measured. Methanol was used a negative control. The percentage of inhibition and the $IC_{50}$ was determined following the same methods as described in Section 2.6.1.

## 3. Results and Discussion

### 3.1. Structures of the Isolated Compounds

The isolated compounds were identified as obliquumol (12-*O*-acetylptaeroxylinol or ptaeroxylinolacetate) [30,31], mixture of lupeol and a minor triterpenoid, possibly β-amyrin [32,33], and *O*-methylalloptaeroxylin (Appendix A, Figure A1) [34]. The structures were assigned on the basis of mass spectrometry and NMR spectroscopy data, which were in agreement with those already reported for the compounds [31,35]. The molecular formula of 12-*O*-acetylptaeroxylinol was confirmed to be $C_{17}H_{16}O_6$ from the HRESIMS data, which showed a protonated molecular ion peak at *m/z* 317.1027 [M + H]$^+$ (Cald For $C_{17}H_{17}O_6$, 317.1025). The MS of the mixture of lupeol and β-amyrin showed the molecular ion peaks at *m/z* 427.4 [M + H]$^+$ and 425.3 [M − H]$^-$, in agreement with the proposed structures. Several studies on *O*-methylalloptaeroxylin report $^1$H NMR data only [34,35]. Therefore, the $^1$H and $^{13}$C NMR data of *O*-methylalloptaeroxylin is provided in Table A1 (Appendix B). The assignment was based on 1D and 2D NMR data. The $^1$H NMR spectrum showed two one-proton singlets at $\delta_H$ 5.99 (H-3) and 6.26 (H-6), and two three-proton singlets at $\delta_H$ 2.28 (CH$_3$-2) and 3.91 (OCH$_3$-5). The signals of the dimethylpyran unit appeared as two doublets at $\delta_H$ 6.69 (H-1′) and 5.56 (H-2′), and a singlet integrating for six protons at $\delta_H$ 1.47 ((CH$_3$)$_2$-3′). The $^{13}$C and dept-135 NMR spectra showed signals of the 5-methoxy and 2-methyl substituents at $\delta_C$ 56.3 and 19.7, respectively. Those of the chromone core appeared at $\delta_C$ 162.6 (C-2), 111.8 (CH-3), 177.6 (C = O, C-4), 108.4 (C-4a), 160.6 (C-5), 96.3 (CH-6), 157.6 (C-7), 102.3 (C-8), 154.2 (C-8a), and the signals of the dimethylpyran scaffold resonated at $\delta_C$ 115.2 (CH-1′), 127.3 (CH-2′), 77.9 (C-3′) and 28.2 ((CH$_3$)$_2$-3′). The molecular formula of the compound was confirmed to be $C_{16}H_{16}O_4$ from the HRESIMS data, which showed a protonated molecular ion peak at *m/z* 273.1140 (Cald for $C_{16}H_{17}O_4$, 273.1127).

### 3.2. Cytotoxicity

In categorizing cytotoxicity of plant extracts, the US National Cancer Institute Guidelines consider extracts to have noteworthy in vitro anti-proliferative activity against cancer cells if 50% inhibitory concentration (IC$_{50}$) value is less than 20 μg/mL, extracts with IC$_{50}$ ranging from 20 μg/mL to 50 μg/mL are considered moderately toxic while those from 50 μg/mL to 200 μg/mL are less toxic and IC$_{50}$ above 200 μg/mL are non-toxic [9]. The acetone crude extracts were more cytotoxic than water extracts and had substantial antiproliferative activity with IC$_{50}$ values ranging from 8 to 374 μg/mL (Table 1). Chloroform fractions were relatively non-toxic to Vero cells with IC$_{50}$ values as high as 284 μg/mL and also had moderate toxicity with IC$_{50}$ values of 33 μg/mL on HepG2 cells. Acetone extracts had better antiproliferative activity compared to aqueous extracts and fractions. Liver and cervical cancer cells showed susceptibility against Hatfield and Walter Sisulu plant acetone extracts while breast cancer cells were susceptible against SANBI acetone extracts. However, SANBI acetone extracts were toxic against normal kidney cells, which questions their safety. These results also corroborated previous studies conducted on Vero cells [19]. The aqueous extracts had lower cytotoxic activity against normal cell lines tested. Similar findings were seen in other studies, which is encouraging, because traditional medicine made from plants is typically prepared as decoctions, infusions, and tinctures made primarily from water [36]. However, it is important to note that all the aqueous extracts tested in the study were also not toxic to all the cancer cell lines tested in the study. The three isolated bioactive compounds were not toxic to the normal cells and the cancer cell lines tested in the study. Therefore, the isolated compounds from the non-polar fraction appear to not be those responsible for the low toxicity observed in both the acetone extracts and the chloroform fraction from which all the compounds were isolated.

**Table 1.** Cytotoxicity (IC$_{50}$ in μg/mL) and selectivity index (SI) of the extracts, fractions, and isolated compounds from *P. obliquum*.

| Extracts | IC$_{50}$ (μg/mL) | | | | | | | | |
|---|---|---|---|---|---|---|---|---|---|
| | MCF7 | SI | HEPG2 | SI | A549 | SI | HELA | SI | VERO |
| **Walter Sisulu** | | | | | | | | | |
| Acetone | 197.3 ± 26.5 | 0.6 | **14.5 ± 0.2** | **8.6** | 147.4 ± 9.6 | 0.8 | 87.2 ± 9.6 | 1.4 | 126.1 ± 4.5 |
| H$_2$O (cold) | 487.8 ± 11.9 | 0.9 | 832.1 ± 42.1 | 0.5 | 353.1 ± 59.5 | 1.3 | 946.6 ± 104.9 | 0.5 | 449.5 ± 0,8 |
| H$_2$O (hot) | 418.7 ± 175.4 | 0.5 | 455.8 ± 24.1 | 0.5 | 830 ± 60.9 | 0.3 | 911.6 ± 56.6 | 0.2 | 214.3 ± 15.1 |
| **UP Hatfield** | | | | | | | | | |
| Acetone | 194.7 ± 27.2 | 0.6 | **8.6 ± 0.8** | **14.2** | 64.1 ± 20.4 | 1.9 | **34.8 ± 6.9** | **3.5** | 122.1 ± 6.1 |
| H$_2$O (cold) | >1000 | 0.3 | 754.6 ± 22.2 | 0.7 | >1000 | 0.2 | >1000 | 0.4 | 535.3 ± 20.5 |
| H$_2$O (hot) | 666.7 ± 109.6 | 1.5 | 372.2 ± 8.3 | 2.74 | 490.8 ± 117.1 | 2.1 | >1000 | 1 | >1000 |
| **Sanbi** | | | | | | | | | |
| Acetone | 23.3 ± 6.6 | 0.7 | 85.8 ± 6.8 | 0.2 | 166.9 ± 20.6 | 0.0 | 99.6 ± 4.9 | 0.2 | **16.1 ± 0.7** |
| H$_2$O (cold) | 764.1 ± 18.9 | 0.6 | >1000 | 0.5 | 188.7 ± 12.3 | 2.6 | 820.4 ± 169.8 | 0.6 | 485.9 ± 121.9 |
| H$_2$O (hot) | >1000 | 0.4 | 607 ± 146.9 | 0.8 | >1000 | 0.2 | 694.5 ± 61 | 0.7 | 464.4 ± 90.3 |
| CHCl$_3$ fraction | 357.6 ± 11.9 | 0.0 | 213.3 ± 18.8 | 0.1 | 129.4 ± 25.4 | 0.2 | 67.2 ± 5.6 | 0.5 | 32.6 ± 3.1 |
| Hexane fraction | 167.5 ± 34.6 | 0.2 | 295.4 ± 18.1 | 0.1 | 250.9 ± 34.9 | 0.1 | 971.6 ± 81.3 | 0.0 | 37.8 ± 4.2 |
| **Nelspruit** | | | | | | | | | |
| Acetone | 269.8 ± 33.2 | 0.4 | 248.4 ± 38.9 | 0.40 | 374.7 ± 8.4 | 0.27 | >1000 | 0.07 | 100.3 ± 0.8 |
| H$_2$O (cold) | >1000 | 0.7 | 246 ± 4.6 | 3.91 | 961.5 ± 137.1 | 1.00 | >1000 | 0.71 | 961.5 ± 19.2 |
| H$_2$O (hot) | 658 ± 162.1 | 0.5 | 550.9 ± 70.4 | 0.59 | 136.6 ± 17.8 | 2.4 | >1000 | 0.18 | 322.5 ± 85.9 |
| CHCl$_3$ fraction | 284.2 ± 38.4 | 1.0 | **33.5 ± 3** | **8.5** | 218.9 ± 9 | 1.3 | 824.5 ± 139.1 | 0.34 | 284.2 ± 68.1 |
| Hexane fraction | 189.2 ± 12.8 | 1.1 | 312.4 ± 16.7 | 0.7 | 180.4 ± 33.4 | 1.1 | 153.1 ± 2 | 1.3 | 203.2 ± 2.5 |
| Obliquumol | 454.2 ± 57 | 0.7 | 52.7 ± 4.8 | 6 | 192.7 ± 1.6 | 1.6 | 188.5 ± 1.6 | 1.7 | 314.8 ± 24.1 |
| Lupeol & β-amyrin | 167.8 ± 6.7 | 0.7 | 122.6 ± 1.8 | 1 | 247.1 ± 49.1 | 0.5 | 247.1 ± 2.7 | 0.5 | 122.6 ± 5.5 |
| O-methylalloptaeroxylin | 248.2 ± 0.1 | 0.6 | 364.4 ± 15.7 | 0.4 | 279.8 ± 57.6 | 0.5 | 212.7 ± 1.8 | 0.7 | 151.5 ± 38.7 |
| Doxorubicin | 0.18 ± 0.01 | 55 | 2.73 ± 0.36 | 3.6 | 1.6 ± 0.04 | 6.3 | 1.6 ± 0.07 | 6.3 | 9.9 ± 1.3 |

The main objective of cancer therapy is to use compounds that can specifically target cancer cells without toxicity against normal cells. Thus, the selective toxicity of extracts, fractions or compounds against cancer cells must be considered during discovery of leads for cancer treatment [37]. We therefore determined if *P. obliquum* acetone extracts, fractions and isolated compounds had selective activity to cancer cells. Tested samples with SI > 2 were considered to have selective toxicity against the tested cancer cell line [38,39]. Acetone extract results also had selective cytotoxic activity against HepG2 and HeLa cancer cell lines with selective index values as high as 14 (Table 1). The Hatfield acetone extract showed the highest cytotoxic activity with $IC_{50}$ of 8.4 μg/mL against HepG2 cells, and had the highest SI value of 14, which means that the extracts were approximately 7 times more toxic to cancer cells than normal Vero cells.

The plant extracts, which showed significant activity against the tested cancer cells, were prepared from organic solvent (acetone), and similar results have been observed in other scientific studies where organic solvents were found to possess more antiproliferative activity than aqueous extracts [40,41]. The type of solvent used for extraction clearly plays a crucial role as it determines the class and polarity of compounds which may be isolated. The extractant used influences the biological activity of *P. obliquum* extract, particularly against the normal and cancer cell lines used in the study.

A wide variety of secondary metabolites are produced by plants, typically as a coping mechanism against attacks from microbes, insects, viruses, herbivores, and other plants [19]. Due to seasonal shifts and geographic location, a plant's chemical composition can alter over time [42]. Geographical location appears to have affected the antiproliferative activity and perhaps phytochemical composition of *P. obliquum*. It was interesting to note that SANBI acetone leaf extracts showed some toxicity against normal cell lines tested, while other acetone leaf extracts collected from different geographical locations were all less toxic. It is likely that the plant leaf material collected from SANBI had a higher concentration of toxic compounds compared to plant material collected from other different geographical locations. Moreover, two acetone extracts from Hatfield and Walter Sisulu, which had the best anticancer activity against HepG2 and HeLa cells, were less toxic to the normal cell lines. This further indicates that there was a difference in the phytochemical concentrations from this plant species based on the geographical location.

### 3.3. Morphology of HepG2 and HeLa Cells

Figures 1 and 2 show modifications in the morphology of HepG2 and HeLa cells caused by *P. obliquum* acetone leaf extracts. Generally, both types of cancer cells were in a scattered pattern, with most of the cells dead and appearing as floating, rounded cells when compared to the adherent spindle-shaped live cells. As expected, a concentration-dependent effect was observed in the morphology of both cell lines tested in the study. Significant cell death and morphological alterations were observed more on HepG2 cells as compared to HeLa cells after 48 h treatment with 10 and 25 μg/mL Hatfield and Walter Sisulu acetone extracts. Compared to the control, both the HepG2 and HeLa cells lost their typical shape and morphology, became rounded and lost their adherence capacity after the exposure of 10 and 25 μg/mL of the *P. obliquum* acetone leaf extracts. It is evident that at the tested concentrations, the acetone leaf extracts of *P. obliquum* are effectively cytotoxic and alter the cell morphology of HepG2 and HeLa cells.

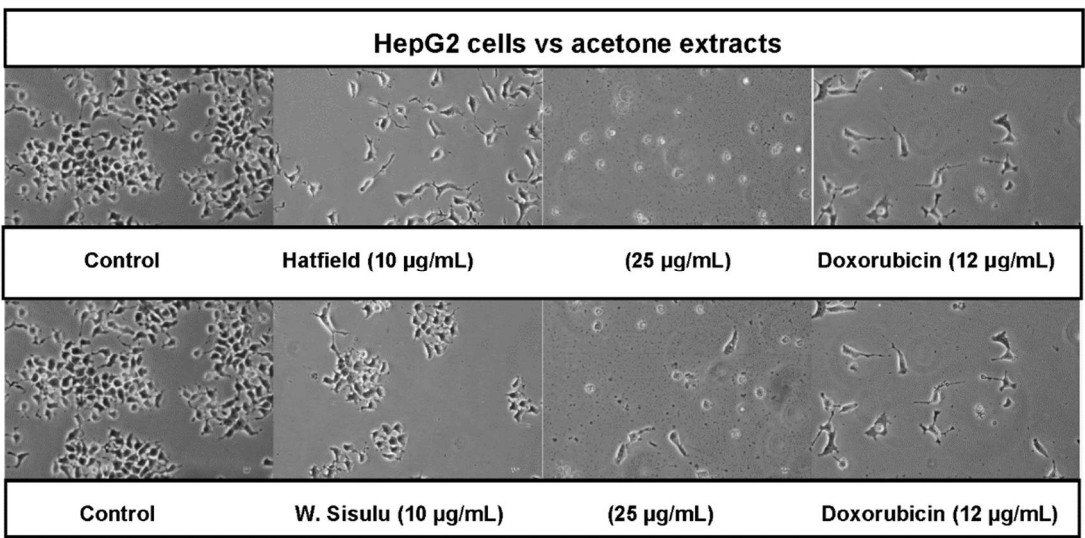

**Figure 1.** Morphological characterization of HepG2 cell lines treated with 10 and 25 µg/mL of Hatfield and Walter Sisulu acetone crude extracts, and 12 µg/mL doxorubicin for 48 h.

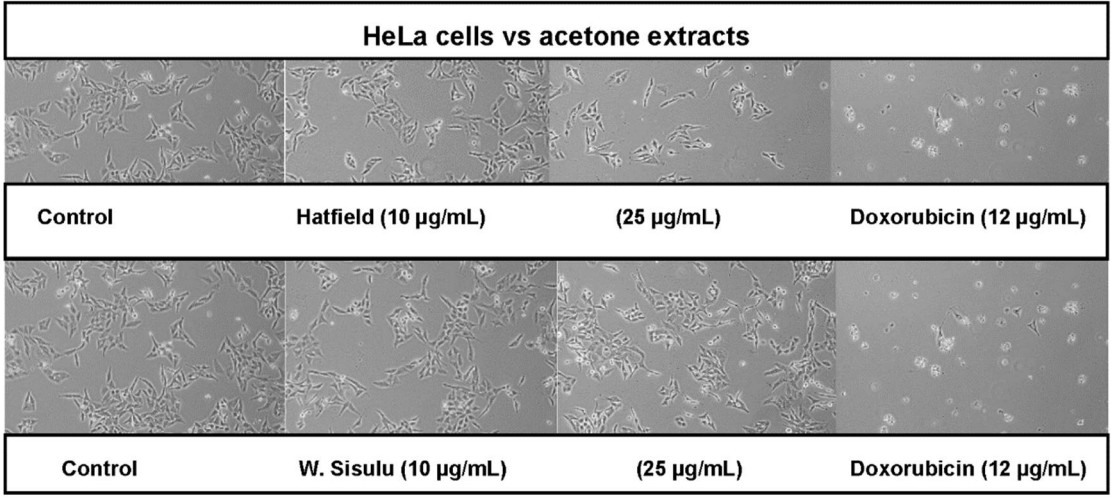

**Figure 2.** Morphological characterization of HeLa cell line treated with 10 and 25 µg/mL of Hatfield and Walter Sisulu acetone crude extracts, and 12 µg/mL doxorubicin for 48 h.

*3.4. Antioxidant Activity of P. obliquum Extracts and Fractions*

A molecule or atom that has one or more unpaired electrons and may exist on its own is referred to as a free radical. The hydroxyl free radical, superoxide free radical anion, lipid peroxyl, lipid peroxide, and lipid alkoxyl are a few examples of free radicals. Radical derivatives such as singlet oxygen and hydrogen peroxide are known as reactive oxygen species (ROS) [43]. The initial line of defense against oxidative stress and damage brought on by free radicals is comprised of antioxidant enzymes. There is a possibility that a disease such as cancer could emerge when there is an imbalance between oxidative stress and antioxidant enzymes [44]. The antioxidant potential of the extracts and fractions from *P obliquum* collected from different geographical locations was determined using the DPPH and ABTS assays, as shown in Table 2. These assays are some of the most commonly used radical scavenging assays methods for determining antioxidant efficacy of natural products and functional food materials because they are rapid, inexpensive and reproducible [45]. It has been proposed that the radical scavenging activity of extracts with $IC_{50} < 100$ µg/mL reflects good antioxidant potential, and extracts with $IC_{50} < 50$ µg/mL are considered to be potent antioxidant agents [46]. The aqueous extract was the most active extract with the

lowest IC$_{50}$ of 21.5 µg/mL on ABTS. The aqueous extract had better scavenging activity against ABTS. Generally, the acetone extract and the two non-polar fractions had less scavenging activity in all the antioxidant methods used. Oxidative stress plays a role in various clinical conditions, including cancer. However, given that the acetone extracts of *P. obliquum* have extremely little antioxidant activity in the methods utilized in this investigation, it indicates that the antiproliferative activity exerted by these extracts may not have been caused by oxidative stress.

**Table 2.** Antioxidant activities (IC$_{50}$ µg/mL) of the *P. obliquum* acetone extracts, aqueous extracts, and fractions.

| | DPPH | ABTS |
|---|---|---|
| **Extracts** | **IC$_{50}$ µg/mL** | |
| **Walter Sisulu** | | |
| Acetone | 269.1 ± 4.6 | 251.2 ± 50 |
| H$_2$O (cold) | 138.3 ± 17.5 | **37.5 ± 10** |
| H$_2$O (hot) | **43.4 ± 6.1** | **21.5 ± 0.2** |
| **UP Hatfield** | | |
| Acetone | 150.6 ± 12 | 178.4 ± 17 |
| H$_2$O (cold) | 140 ± 9.3 | 86.1 ± 1.5 |
| H$_2$O (hot) | 85.4 ± 6.6 | 59 ± 0.2 |
| **Sanbi** | | |
| Acetone | 275.5 ± 8.9 | 318.1 ± 19.2 |
| H$_2$O (cold) | 75.7 ± 2.5 | **43 ± 1.6** |
| H$_2$O (hot) | **46.1 ± 9.5** | **29.1 ± 0** |
| CHCl$_3$ fraction | 423.5 ± 54.3 | 240.4 ± 28.8 |
| Hexane fraction | 418.5 ± 9.6 | 143.7 ± 3.3 |
| **Nelspruit** | | |
| Acetone | 333.2 ± 24.9 | 268 ± 29.9 |
| H$_2$O (cold) | 62.1 ± 6.1 | **36.6 ± 0.6** |
| H$_2$O (hot) | 83.1 ± 3.2 | 56.4 ± 4.4 |
| CHCl$_3$ fraction | 387.4 ± 27.3 | 214.2 ± 13.1 |
| Hexane fraction | 236.5 ± 42.1 | 180.2 ± 2.7 |
| **Trolox** | 2.4 ± 0.8 | 1.6 ± 0.0 |
| **Ascorbic Acid** | 2.6 ± 0.2 | 1.4 ± 0.2 |

## 4. Conclusions

Since there is growing evidence that some phytochemicals may have anticancer properties, plants present a useful source of potential alternative anticancer treatments. Many phytochemicals have a significant anticancer potential and could be used in drug development. Traditional medicinal plants have found favour in modern medicine as possible sources for new therapeutic medications.

Natural products have historically led to the discovery of numerous innovative anticancer medications. Many of these products have demonstrated preliminary anticancer action in vitro, as evidenced by their ability to be cytotoxic or antiproliferative, as well as by their impact on mechanisms involved in cancer cell growth. The major goals of studying crude plant extracts are to either isolate bioactive compounds for use directly as medications or to find bioactive compounds that can be utilized as lead ingredients in the development of semi-synthetic pharmaceuticals, thus it is imperative to isolate the cytotoxic compounds in *P. obliquum*. These cytotoxic natural products may have a significant role in treating selected cancers by working in synergy with conventional chemotherapeutic drugs, possibly by reducing toxicity while improving their efficacy. Antiproliferative agents such as *P. obliquum* acetone extracts, that can induce selective cytotoxicity against cancer cell lines without causing much harm to normal cells, are highly desirable for therapeutic purposes and may be considered in the development of novel cancer chemotherapeutic drugs.

**Author Contributions:** Conceptualization, E.T.K., T.E.R., J.N.E. and L.J.M..; methodology, E.T.K., S.M.N. and M.A.S.; formal analysis, E.T.K., S.M.N. and M.A.S.; data curation, E.T.K.; writing—original draft preparation, E.T.K.; writing—review and editing, S.M.N., M.A.S., J.N.E., T.E.R., L.J.M.; supervision, J.N.E., T.E.R., L.J.M.; funding acquisition, L.J.M. All authors have read and agreed to the published version of the manuscript.

**Funding:** The authors are grateful to the University of Pretoria, National Research Foundation (NRF, grant number 105993) and the Health and Welfare Sector Education and Training Authority (HWSETA) for financial assistance.

**Institutional Review Board Statement:** The study was approved by the Faculty of Veterinary Sciences Research Ethics Committee, University of Pretoria (REC235-19 on 07/02/2020).

**Data Availability Statement:** All data applicable to the article are presented here.

**Acknowledgments:** The Lowveld National Botanical Garden, Walter Sisulu Botanical Garden, Pretoria National Botanical Garden and the University of Pretoria are thanked for allowing collection of plant material.

**Conflicts of Interest:** The authors declare no conflict of interest. The funders had no role in the design of the study; in the collection, analyses, or interpretation of data; in the writing of the manuscript; or in the decision to publish the results.

## Appendix A

**Figure A1.** Structures of isolated compounds from *P. obliquum* leaves.

## Appendix B

**Table A1.** $^1$H NMR (500 MHz) and $^{13}$C NMR (125 MHz) of *O*-methylalloptaeroxylin in CDCl$_3$.

| Position | $\delta_{\text{C}}$ | $\delta_{\text{H}}$ (*J* in Hz) |
|----------|---------|---------------------|
| 2 | 162.6 | |
| 3 | 111.8 | 5.99, s |
| 4 | 177.6 | |
| 4a | 108.4 | |
| 5 | 160.6 | |
| 6 | 96.3 | 6.26, s |
| 7 | 157.6 | |
| 8 | 102.3 | |
| 8a | 154.2 | |
| CH$_3$-2 | 19.7 | 2.28, s |
| OMe-5 | 56.3 | 3.91, s |
| 1' | 115.2 | 6.69, d (10.4) |
| 2' | 127.3 | 5.56, d (10.4) |
| 3' | 77.9 | |
| (CH$_3$)$_2$-3' | 28.2 | 1.47, s |

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
