# Peer review of "In Vitro Antiproliferative Activity of Ptaeroxylon obliquum Leaf Extracts, Fractions and Isolated Compounds on Several Cancer Cell Lines"

_applsci, doi:10.3390/app122111004_

Round 1

Reviewer 1 Report

The manuscript focused the study of in vitro bioactivities of Ptaeroxylon obliquum leaf extract such as anticancer and antioxidant activities. The experiment protocol is provided in details and the results are supported by experimental data. The reviewer highly recommends the publication of the manuscript after considering making a few minor revisions as follows

1) Can the author give some more information on herbal medicine use history of  Ptaeroxylon obliquum leaf if there is any?

2) When isolated compounds such as compound 4 and compound 5 using column chromatography, have the authors use any detectors such UV or fluorescence to monitor the components in different elution fractions? If so, please provide the detector type.

3) For section 3.1. , can the author provide more information about mass spectrometry or HPLC analysis of the chemical components in the isolated compounds?

Author Response

The manuscript focused the study of in vitro bioactivities of Ptaeroxylon obliquum leaf extract such as anticancer and antioxidant activities. The experiment protocol is provided in details and the results are supported by experimental data. The reviewer highly recommends the publication of the manuscript after considering making a few minor revisions as follows

1) Can the author give some more information on herbal medicine use history of Ptaeroxylon obliquum leaf if there is any?

Response: More information on the traditional uses of P. obliquum was added in the Introduction, pages 3 and 4.

2) When isolated compounds such as compound 4 and compound 5 using column chromatography, have the authors use any detectors such UV or fluorescence to monitor the components in different elution fractions? If so, please provide the detector type.

Response: Thank you for noting this. The information on compounds 4 and 5 has been deleted as it was mistakenly included in the manuscript.

3) For section 3.1., can the author provide more information about mass spectrometry or HPLC analysis of the chemical components in the isolated compounds?

Response: The mass spectrometry data of the isolated compounds is now explained on page 8, section 3.1.

Reviewer 2 Report

The authors described the antioxidant and antiproliferative activity of Ptaeroxylon obliquum organic/ aqueous extracts as well as some isolated compounds. Little novelty is introduced in this study as well as the results.

Some general comments are addressed below:

References should be reviewed and actualized.

For example: Ramadwa TE, Awouafack MD, Sonopo MS, Eloff JN. Antibacterial and Antimycobacterial Activity of Crude Extracts, Fractions, and Isolated Compounds From Leaves of Sneezewood, Ptaeroxylon obliquum (Rutaceae). Natural Product Communications. 2019;14(11). doi:10.1177/1934578X19872927

How do you prepare the sample added to the cancer cells? It is not clear in the manuscript.

Please verify the designation compound 4 and compound 5, it is not identified in the manuscript.

Please verify. Reference 35 is the same of 36, only the authors is different.

Please verify the 12-O-acetylptaeroxylinol structure.

The authors could provide the NMR data of O-methylalloptaeroxylin in the appendix.

Please correct. The authors present the morphology effect of two concentration (10 and 25 μg/mL) but the manuscript present 10-25 μg/mL, seen like the authors use a range of concentration.

Please correct figure 1. Introduce the treatment time and correct the cells treatments.

Author Response

The authors described the antioxidant and antiproliferative activity of Ptaeroxylon obliquum organic/ aqueous extracts as well as some isolated compounds. Little novelty is introduced in this study as well as the results.

Some general comments are addressed below:

References should be reviewed and actualized.

For example: Ramadwa TE, Awouafack MD, Sonopo MS, Eloff JN. Antibacterial and Antimycobacterial Activity of Crude Extracts, Fractions, and Isolated Compounds From Leaves of Sneezewood, Ptaeroxylon obliquum (Rutaceae). Natural Product Communications. 2019;14(11). doi:10.1177/1934578X19872927

Response: Thank you, this has been addressed.

How do you prepare the sample added to the cancer cells? It is not clear in the manuscript.

Response: The preparation procedure was added on page 6.

Please verify the designation compound 4 and compound 5, it is not identified in the manuscript.

Response: The statement has been deleted since the proton and carbon NMR of those compounds were inconclusive so all data on isolation of these compounds was deleted.

Please verify. Reference 35 is the same of 36, only the authors is different.

Response: Thank you for noting this - it was an error so the reference was removed.

Please verify the 12-O-acetylptaeroxylinol structure.

Response: The structure of 12-O-acetylptaeroxylinol has been redrawn correctly (Figure A1).

The authors could provide the NMR data of O-methylalloptaeroxylin in the appendix.

Response: The NMR data table was moved to the appendix as suggested

Please correct. The authors present the morphology effect of two concentration (10 and 25 μg/mL) but the manuscript present 10-25 μg/mL, seen like the authors use a range of concentration.

Response: The concentration was rewritten to “10 and 25 μg/mL”.

Please correct figure 1. Introduce the treatment time and correct the cells treatments.

Response: The time and treatment were amended as requested.